# Prevalence of Chronic Obstructive Pulmonary Disease and Asthma in the Community of Pathumthani, Thailand

**DOI:** 10.3390/diseases13050130

**Published:** 2025-04-23

**Authors:** Narongkorn Saiphoklang, Pitchayapa Ruchiwit, Apichart Kanitsap, Pichaya Tantiyavarong, Pasitpon Vatcharavongvan, Srimuang Palungrit, Kanyada Leelasittikul, Apiwat Pugongchai, Orapan Poachanukoon

**Affiliations:** 1Department of Internal Medicine, Faculty of Medicine, Thammasat University, Pathum Thani 12120, Thailand; toon.pitchayapa@gmail.com (P.R.); apidoctor@hotmail.com (A.K.); pichaya_t@tu.ac.th (P.T.); 2Medical Diagnostics Unit, Thammasat University Hospital, Pathum Thani 12120, Thailand; lee.kanyada@gmail.com (K.L.); pu.apiwat@gmail.com (A.P.); 3Center of Excellence for Allergy, Asthma and Pulmonary Diseases, Thammasat University Hospital, Pathum Thani 12120, Thailand; orapanpoachanukoon@yahoo.com; 4Department of Clinical Epidemiology, Faculty of Medicine, Thammasat University, Pathum Thani 12120, Thailand; 5Department of Community Medicine and Family Medicine, Faculty of Medicine, Thammasat University, Pathum Thani 12120, Thailand; pasitpon@tu.ac.th (P.V.); srimuangpa@gmail.com (S.P.); 6Department of Pediatrics, Faculty of Medicine, Thammasat University, Pathum Thani 12120, Thailand

**Keywords:** asthma, chronic obstructive pulmonary disease, COPD, pulmonary function, respiratory symptoms

## Abstract

**Background**: Airway diseases, particularly asthma and chronic obstructive pulmonary disease (COPD), pose significant respiratory problems. The prevalence and risk factors of these diseases among community dwellers vary geographically and because of underdiagnosis. This study aims to determine the prevalence and factors associated with these diseases in a provincial-metropolitan area in Thailand. **Methods**: A cross-sectional study was conducted between April 2023 and November 2023 on individuals aged 18 years or older residing in Pathumthani, Thailand. Data on demographics, pre-existing diseases, respiratory symptoms, and pulmonary functions assessed by spirometry, including forced vital capacity (FVC), forced expiratory volume in one second (FEV_1_), and bronchodilator responsiveness (BDR), were collected. COPD was defined as having respiratory symptoms, a risk factor, and post-bronchodilator FEV_1_/FVC < 70%. Asthma was defined as having respiratory symptoms and a positive bronchodilator responsiveness. **Results**: A total of 1014 subjects (71.7% female) were included, with a mean age of 56.6 years. The smoking history was 10.4% (13.4 pack-years). Common symptoms included cough (18.4%), sputum production (14.5%), and dyspnea (10.0%). COPD was found in 8.3%, while asthma was found in 10.3%. Logistic regression analysis indicated that these diseases were significantly associated with older age (odds ratio [OR] 1.023; 95% confidence interval [CI] 1.007–1.039 for every 1-year increase in age), smoking (OR 2.247; 95% CI 1.068–4.728), heart disease (OR 2.709; 95% CI 1.250–5.873), wheezing (OR 3.128; 95% CI 1.109–8.824), runny nose (OR 1.911; 95% CI 1.050–3.477), and previous treatment for dyspnea (OR 6.749, 95% CI 3.670–12.409). **Conclusions**: COPD and asthma were relatively prevalent in our study. Being elderly, smoking, having heart disease, and experiencing any respiratory symptoms with a history of treatment are crucial indicators for these airway diseases. Pulmonary function testing might be needed for active surveillance to detect these respiratory diseases in the community.

## 1. Introduction

Chronic obstructive pulmonary disease (COPD) and asthma are common airway diseases and significant public health problems globally [1]. Currently, several countries, including Thailand, face numerous air pollution problems due to traffic fumes, incinerated rubbish, industrial emissions, agricultural burning, and forest fires. This pollution contains fine particulate matter less than 10 microns (PM10) in diameter and even smaller than 2.5 microns (PM2.5), which contributes to the development and exacerbation of chronic airway diseases, especially COPD and asthma [2,3,4]. Deaths from COPD are eight times higher than deaths from asthma [5].

COPD is a leading cause of death globally [6]. The disease is commonly found in the elderly. Cigarette smoking is an important risk factor, and other factors, such as exposure to indoor and outdoor air pollution, occupational hazards, and infections, are also important [7]. Global prevalence of COPD is approximately 12.6% [8]. Men have a higher prevalence of COPD compared to women (15.5% vs. 8.8%) [8]. Patients often experience respiratory symptoms, limited daily activities, and reduced lung function [9]. Worsening symptoms or exacerbations of COPD can result in hospitalization and impair a patient’s quality of life [6,7,10]. Early diagnosis and intervention can prevent disease progression and exacerbation and help to maintain a good quality of life [6,11].

Asthma is a common chronic inflammatory airway disease. The global prevalence of asthma has been estimated to be 5.4%, but in 2019 it was approximately 9.8% [12]. Patients usually have respiratory symptoms triggered by allergens, exercise, or respiratory infections. Acute exacerbations can lead to morbidity and mortality [13].

The prevalence and risk factors of these diseases vary geographically. This study provides novel insights by offering updated epidemiological data on COPD and asthma in Thailand. We hypothesize that the prevalence of COPD and asthma in a provincial-metropolitan area in Thailand is significantly associated with smoking and air pollution exposure. We expect to find a relatively high prevalence of both COPD and asthma in the study area, with cigarette smoking and environmental pollutants as the primary associated risk factors. This study aimed to determine the prevalence and factors associated with COPD and asthma in a provincial-metropolitan area in Thailand.

## 2. Materials and Methods

### 2.1. Study Design and Participants

A cross-sectional study was conducted on people residing in Pathumthani, located 40 km from Bangkok, Thailand, between April 2023 and November 2023. Data were randomly collected from 1014 people in seven districts of Pathumthani. Individuals aged 18 years or older were included. Exclusion criteria included inability to perform spirometry, active respiratory symptoms such as severe cough and dyspnea, active respiratory infections such as COVID-19, common cold, or pulmonary tuberculosis, recent myocardial infarction, blood pressure higher than 180/100 mmHg, and resting heart rate greater than 120 beats per minute.

The study protocol was approved by the Human Research Ethics Committee of Thammasat University (Medicine) (IRB No. MTU-EC-IM-4-235/65, COA No. 015/2023 Date of approval: 16 January 2023). All participants provided written informed consent. This study was prospectively registered with Thaiclinicaltrials.org with the number TCTR20230711002.

### 2.2. Procedures and Outcomes

Demographic data, pre-existing comorbidities, respiratory symptoms, and lung functions were collected by spirometry, including forced vital capacity (FVC), forced expiratory volume in one second (FEV_1_), forced expiration flow rate at 25–75% of FVC (FEF_25-75_), and bronchodilator responsiveness (BDR). Spirometry was performed according to the international guidelines of the United States and Europe [14,15,16] using a PC-based spirometer (Vyntus SPIRO, Vyaire Medical, Mettawa, IL, USA). To minimize intra-observer variability, all spirometry tests were conducted by the same trained technician using standardized procedures. Quality control was ensured by daily calibration of the spirometer, adherence to guidelines for acceptability and repeatability, and periodic review of test performance by a pulmonologist. Participants were instructed to exhale into the tube forcefully and rapidly, and then to continue exhaling for 15 s or more. FVC, FEV_1_, FEV_1_/FVC, and FEF_25-75_ were reported in liters (L), %predicted, %, or liters per second (L/s). BDR was assessed by inhaling 400 µg of salbutamol and repeating spirometry after 15 min. Predicted values of all spirometry parameters were used according to reference equations of the Global Lung Function Initiative [17]. BDR was defined as increase in FEV_1_ ≥ 12% and ≥200 mL after BDR test [13].

In our study, airway diseases were classified into COPD and asthma. COPD was defined as having respiratory symptoms (such as cough, sputum production, dyspnea, or wheezing), the presence of risk factors, especially smoking ≥10 pack-years or biomass fuel use, and a post-bronchodilator FEV_1_/FVC < 70% [6]. Asthma was defined as having respiratory symptoms (such as wheezing, dyspnea, chest tightness, or cough) and a positive BDR [13].

### 2.3. Statistical Analysis

In a previous study [18], the prevalence of COPD in a Thai population was 7.1%. We hypothesized that the prevalence in our population was the same. Our sample size was calculated to estimate a proportion with a confidence of 80%, a type I error of 5%, and a precision margin of 5%. Therefore, the calculated sample size for estimating COPD prevalence was 102.

Categorical variables were expressed as number (percentage). Continuous variables were expressed as mean ± standard deviation. Chi-squared test was used to compare categorical data between the airway disease groups and normal groups, as well as among the three groups (healthy, COPD, and asthma). Student’s *t*-test was used to compare the means of continuous variables between the two groups. One-way analysis of variance (ANOVA) was used to compare the means of continuous variables among the three groups (healthy, COPD, and asthma). To determine the set of variables associated with the airway disease, we used the logistic regression model with the airway disease as the dependent variable. Independent variables, including age, sex, body mass index, smoking status, occupations, preexisting comorbidities, respiratory symptoms, and previous respiratory treatments, were entered into the regression model if they showed statistical significance in bivariate analysis or were identified as relevant based on prior knowledge to adjust confounders. The backward elimination method was used to select the final model. Adjusted odds ratios (OR) and 95% confidence interval (95% CI) were reported for variables in the model. A two-sided p-value < 0.05 was considered statistically significant. Statistical analyses were performed using SPSS version 26.0 software (IBM Corp., Armonk, NY, USA).

## 3. Results

### 3.1. Participants

A total of 1027 participants were screened. Of these, 1014 were included in the final analysis (71.7% female) (Figure 1). The mean age was 56.6 ± 13.3 years. Current or former smokers comprised 10.4% with an average of 13.4 pack-years. Hypertension (32.5%), hyperlipidemia (26.3%), and diabetes (15.0%) were common comorbidities. Self-reported asthma and COPD were found in 2.7 and 0.9%, respectively. Common respiratory symptoms included cough (18.4%), sputum production (14.5%), and breathlessness (10.0%) (Table 1).

### 3.2. Prevalence of COPD and Asthma

Spirometry data showed FEV_1_/FVC of 82.5%, FEV_1_ of 94.5% predicted, and BDR of 8.9% (Table 2). COPD and asthma were found in 8.3% and 10.3% of patients, respectively (Table 2).

### 3.3. Factors Associated with COPD and Asthma

Compared to participants without airway disease, the airway disease group was significantly older, with a higher proportion of males, smokers, unemployed individuals, hypertension, coronary heart disease, preexisting asthma, preexisting COPD, presence of respiratory symptoms, previous treatment of dyspnea and visits to the emergency department. However, body mass index was lower in the airway disease group (Appendix A). The following Appendix A can be downloaded at: https://www.mdpi.com/article/10.3390/diseases13050130/s1, Appendix A: Univariate analysis for factors associated with airway diseases. Logistic regression analysis indicated that higher age (OR 1.023; 95% CI 1.007–1.039 for every 1-year increase in age), smoking (OR 2.247; 95% CI 1.068–4.728), coronary heart disease (OR 2.709; 95% CI 1.250–5.873), wheezing (OR 3.128; 95% CI 1.109–8.824), runny nose (OR 1.911; 95% CI 1.050–3.477), and previous treatment of dyspnea (OR 6.749, 95% CI 3.670–12.409) were associated with airway diseases (Table 3).

Compared to the patients with asthma, the COPD group of patients had higher proportions of males, smokers, previous diagnoses of COPD, and better pulmonary functions (FVC and FEV_1_), but lower proportions of previous diagnoses of asthma, breathlessness, and BDR (Table 4). One asthmatic patient was misdiagnosed as COPD. Two COPD patients were misdiagnosed as asthma (Table 4).

## 4. Discussion

To the best of your knowledge, this is the largest survey of COPD and asthma in a city in Central Thailand in the past two decades. The prevalence of COPD and asthma was 8% and 10%, respectively. These airway diseases were associated with older age, smoking history, coronary heart disease, wheezing, runny nose, and previous treatment of dyspnea. This study offers novel insights by providing updated epidemiologic data on the prevalence and risk factors of COPD and asthma in a provincial-metropolitan community in Pathumthani, Thailand—a region that has been underrepresented in previous research.

The clinical characteristics and pulmonary function profiles observed in our study both align with and diverge from findings reported in previous studies conducted in different populations and settings. In our cohort of 1014 individuals, the prevalence of COPD and asthma was 8.3% and 10.3%, respectively. These figures are comparable to the global estimates reported in studies such as those by Adeloye, et al. (2015) [19] and To, et al. (2012) [20], which demonstrated a COPD prevalence ranging from 8.4% to 15.0% and an asthma prevalence ranging from 1.0% to 21.5%, depending on age and geographic region. Among our COPD patients, 50.0% were male, with a mean age of 60.9 years. This contrasts with the ECLIPSE study, which reported approximately 65% male participants with a mean age of 63 years [21]. In our asthma patients, 75% were female, with a mean age of 61.2 years. Similarly, studies of European populations on adult-onset and late-onset asthma often report a female predominance (54–71%), and typically include individuals in middle to older age groups (31–61 years) [22].

The spirometry findings in our COPD patients showed an average FEV_1_/FVC ratio of 68% and FEV_1_ of 85% of the predicted values. These values indicate higher lung function than those reported in the ECLIPSE study, which showed an FEV_1_/FVC ratio of 45% and FEV_1_ at 48% of predicted [21]. This discrepancy may be due to our study representing early, community-based screening, whereas the ECLIPSE study focused on patients already receiving care in healthcare centers.

Our study found that the prevalence of COPD was slightly higher than in previous studies in Bangkok in 2002 and Chiang Mai Province in 2015, which found 4–7% prevalence among Thai people [18,23]. This difference might be attributed to increased air pollution, especially PM2.5, rather than to increased cigarette smoking. PM2.5 is a significant risk factor for COPD [24]. Long-term exposure to PM2.5 has been associated with increased incidence and prevalence of the disease [25,26]. Ambient concentrations of PM2.5 are strongly correlated with reduced pulmonary function and greater emphysema, even at relatively low levels [27,28]. Globally, the number of COPD-related deaths and disability-adjusted life years (DALYs) attributable to ambient PM2.5 increased by more than 90% between 1990 and 2019 [29]. In Thailand, smoking decreased over the past decade, as shown in a study conducted by Aungkulanon S, et al., which found an overall reduction in smoking from 23% in 2003 to 19% in 2017 [30].

In addition, the prevalence of asthma in our study was higher than in a study by Boonsawat W, et al., which found the prevalence of asthma in Thai adults in 2004 to be 7% [31]. This difference may be explained by longer life expectancy in Thailand, improved awareness and diagnosis of asthma among physicians, and rising levels of air pollution [3,32]. Furthermore, a study conducted in a northern Thai city reported a 5.5% prevalence of chronic airflow obstruction among villagers [33]. In that study, villagers had significantly lower FEV_1_/FVC ratios compared to government employees (98% vs. 100%; *p* = 0.04). However, farming activities and pesticide exposure were not found to be associated with reduced lung function, which aligns with similar occupational findings in our study.

Respiratory diseases, particularly COPD, tend to increase with age [7,34]. The mean age of all participants in our study was 56.6 years, and in the COPD group it was 60.9 years.

Smoking is a well-established risk factor for respiratory diseases, especially COPD [6,35]. Prolonged exposure to particles and gases in cigarette smoke leads to COPD development, which leads to epithelial cell damage and inflammatory cell infiltration in the lung tissue, including macrophages and neutrophils [36]. Former and current smokers comprised 10.4% of all participants in our study, with a mean of 13.4 pack-years.

Coronary heart disease, which is common in COPD patients and can exacerbate COPD and complicate treatment [37,38], was a comorbidity in participants in our study. Our patients had previously been treated for dyspnea and had visited an emergency department in the past year, which suggests exacerbations of COPD or asthma. Frequent COPD exacerbations decrease quality of life and increase healthcare costs [39]. Although cardiovascular (CV) comorbidities are typically more prevalent in patients with COPD [40,41], our study did not observe significant differences between asthma and COPD groups. Patients with either condition may share common CV factors, such as smoking history, systemic inflammation, sedentary lifestyle, or obesity—all of which could contribute to similar rates of CV comorbidity prevalence. Moreover, in older adults, asthma-COPD overlap is common [42], and some patients diagnosed with asthma may in fact have features of COPD or vice versa, potentially blurring distinctions in CV outcomes between the groups.

Interestingly, wheezing and a runny nose were significantly associated with airway diseases in our study, but were not different between patients with COPD and asthma. Runny nose is a common nasal symptom associated with allergic rhinitis (AR) and asthmatic symptoms, as reported by Alrasheedi SM, et al. [43]. AR severity is associated with asthma control, quality of life, and pulmonary function [44]. Some COPD patients may also have nasal symptoms resulting from AR or asthma which complicate COPD management. Wheezing may be found in patients with COPD or asthma.

Our study demonstrated that COPD patients were predominantly male, smokers, had better pulmonary functions, less breathlessness, and less BDR than asthmatic patients. Risk factors for COPD include being male and smoking [6]. Asthmatic patients usually have consistent diagnostic criteria [13]. Surprisingly, COPD patients in our study had better pulmonary functions and a lower rate of breathlessness than asthmatic patients. These findings might result from under-diagnosis of asthma, leading to more respiratory symptoms and airway remodeling, chronic airway obstruction, and poor lung function.

There are limitations to this study. Firstly, we did not to perform chest imaging. Therefore, airway obstruction from causes such as bronchiectasis might result in misdiagnosis. Secondly, we conducted the study in the post-COVID-19 era; some airway diseases might result from prior COVID-19 infection. Thirdly, asthma–COPD overlap might have been found in our study, but distinguishing this condition from pure asthma or COPD was difficult due to uncertain diagnostic criteria and overlapping clinical manifestations. Persistent airflow limitation, indicated by post-bronchodilator FEV_1_/FVC < 70% in our participants, might have been misclassified as COPD, despite the possibility of asthma with fixed airway obstruction not being excluded. This phenomenon may lead to an overestimation of COPD prevalence. In contrast, participants with a strong clinical suspicion of asthma but normal spirometry and negative BDR results may have silent asthma, which also cannot be excluded. Bronchial challenge testing is needed for definitive asthma diagnosis. Moreover, selection bias might have occurred, as only community-dwelling individuals who were available and willing to participate were included. This could have led to either underestimation or overestimation of disease prevalence. In addition, information bias might have arisen from self-reported data, which may have led to misclassification and affected the observed associations between risk factors and disease outcomes. Data on the severity of COPD and asthma, as well as their treatments, were not collected. Lastly, this study did not include long-term follow-up; therefore, we could not assess the clinical progression of airway diseases. Future prospective studies with longer follow-up periods are needed to evaluate changes in lung function and long-term clinical outcomes in these patients.

## 5. Conclusions

COPD and asthma are relatively prevalent in our study. Being elderly, smoking, having heart disease, and experiencing any respiratory symptoms with a history of treatment are crucial indicators for these airway diseases. Pulmonary function testing might be needed for active surveillance to detect these respiratory diseases in the community.

## Figures and Tables

**Figure 1 diseases-13-00130-f001:**
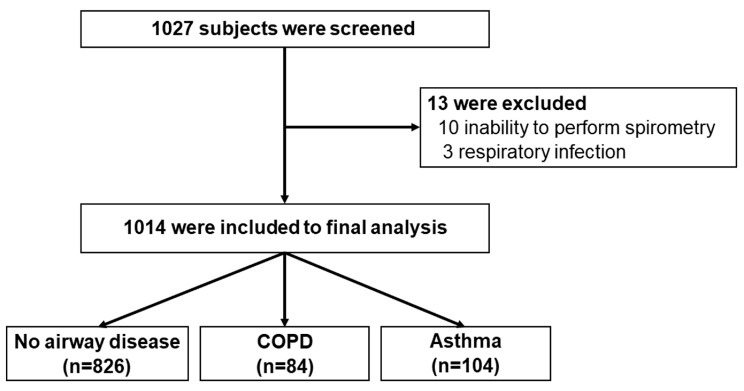
Flowchart of participant recruitment to the study.

**Table 1 diseases-13-00130-t001:** Baseline characteristics of participants.

Characteristics	Total (n = 1014)	No Airway Disease (n = 826)	Airway Disease (n = 188)	*p*-Value
Age, years	56.6 ± 13.3	55.6 ± 12.8	61.0 ± 14.7	<0.001
Female	727 (71.7)	607 (73.5)	120 (63.8)	0.008
Male	287 (28.3)	219 (26.5)	68 (36.2)	0.008
Body mass index, kg/m^2^	25.1 ± 4.6	25.2 ± 4.5	24.5 ± 4.8	0.049
Smoking (current or former)	105 (10.4)	79 (9.6)	26 (13.8)	<0.001
Amount of smoking, pack-years	13.4 ± 16.4	11.7 ± 13.5	17.3 ± 21.1	0.128
Fuel, hours per year	9982.7 ± 10,345.2	9959.3 ± 10,217.3	10,090.6 ± 10,948.3	0.887
**Occupations**				
Government officer	120 (11.8)	104 (12.6)	16 (8.5)	0.118
Farmer	89 (8.8)	73 (8.8)	16 (8.5)	0.886
Merchant	227 (22.4)	201 (24.3)	26 (13.8)	0.002
General worker	150 (14.8)	130 (15.7)	20 (10.6)	0.075
Others	81 (8.0)	67 (8.1)	14 (7.4)	0.823
Unemployed	347 (34.2)	251 (30.4)	96 (51.1)	<0.001
**Preexisting comorbidities**				
Hypertension	330 (32.5)	254 (30.4)	76 (40.4)	0.011
Hyperlipidemia	267 (26.3)	207 (25.1)	60 (31.9)	0.054
Diabetes	152 (15.0)	119 (14.4)	33 (17.6)	0.275
Coronary heart disease	35 (3.5)	18 (2.2)	17 (9.0)	<0.001
Cerebrovascular disease	10 (1.0)	7 (0.8)	3 (1.6)	0.405
Obesity	23 (2.3)	15 (1.8)	8 (4.3)	0.055
Allergic rhinitis	120 (11.8)	96 (11.6)	24 (12.8)	0.661
Asthma	27 (2.7)	0 (0)	27 (14.7)	<0.001
COPD	9 (0.9)	0 (0)	9 (4.8)	<0.001
**Respiratory symptoms**	379 (37.4)	280 (33.9)	99 (52.7)	<0.001
Cough	187 (18.4)	133 (16.1)	54 (28.7)	<0.001
Sputum production	147 (14.5)	110 (13.3)	37 (19.7)	0.025
Breathlessness	101 (10.0)	64 (7.7)	37 (19.7)	<0.001
Wheezes	21 (2.1)	8 (1.0)	13 (6.9)	<0.001
Chest tightness	39 (3.8)	31 (3.8)	8 (4.3)	0.747
Runny nose	69 (6.8)	44 (5.3)	25 (13.3)	<0.001
Nasal obstruction	68 (6.7)	53 (6.4)	15 (8.0)	0.440
Sore throat	34 (3.4)	26 (3.1)	8 (4.3)	0.446
Dyspnea on exertion	42 (4.1)	32 (3.9)	10 (5.3)	0.369
**History of respiratory treatment and cost**				
Previous treatment of dyspnea	59 (5.8)	22 (2.7)	37 (19.7)	<0.001
Prior ED visit in the past year	31 (3.1)	19 (2.3)	12 (6.4)	0.003
Treatment cost, USD in the past year (n = 5)	822 ± 947	680 ± 822	1388	0.582

Data shown as n (%) or mean ± SD. COPD = chronic obstructive pulmonary disease, ED = emergency department, kg = kilogram, m = meter, USD = The United States dollar.

**Table 2 diseases-13-00130-t002:** Lung function data of participants.

Parameters	Total	Healthy	COPD	Asthma	*p*-Value
Number of subjects, n (%)	1014 (100)	826 (81.5)	84 (8.3)	104 (10.3)	NA
FVC, L	2.60 ± 0.71	2.63 ± 0.68	2.86 ± 0.88	2.15 ± 0.67	<0.001
FVC, %predicted	94.2 ± 15.7	94.7 ± 14.3	100.4 ± 19.0	85.7 ± 19.8	<0.001
FVC improvement after BDR test, %	1.2 ± 6.9	−0.1 ± 4.5	−0.2 ± 6.9	12.8 ± 12.1	<0.001
FEV_1_, L	2.11 ± 0.58	2.19 ± 0.55	1.95 ± 0.63	1.65 ± 0.51	<0.001
FEV_1_, %predicted	93.6 ± 16.0	96.0 ± 14.1	84.7 ± 17.1	81.7 ± 20.8	<0.001
FEV_1_ improvement after BDR test, %	3.5 ± 6.1	2.1 ± 3.6	4.3 ± 4.8	15.0 ± 10.9	<0.001
FEV_1_/FVC, %	81.7 ± 7.8	83.5 ± 5.5	68.2 ± 6.3	77.9 ± 11.4	<0.001
FEF_25-75_, L/s	2.11 ± 0.92	2.30 ± 0.86	1.10 ± 0.54	1.46 ± 0.89	<0.001
FEF_25-75_, %predicted	85.2 ± 33.6	92.0 ± 30.6	44.1 ± 15.2	65.0 ± 35.2	<0.001
BDR	90 (8.9)	0 (0)	5 (6.0)	85 (81.7)	<0.001

Data shown as n (%) or mean ± SD. BDR defined as increase in FEV_1_ ≥ 12% and ≥200 mL after BDR test. BDR = bronchodilator response, COPD = chronic obstructive pulmonary disease, FEV_1_ = forced expiratory volume in one second, FVC = forced vital capacity, FEF_25-75_ = forced expiratory flow at 25–75% of FVC, L = liter, mL = milliliter, NA = not applicable, s = second.

**Table 3 diseases-13-00130-t003:** Logistic regression analysis for factors associated with airway diseases.

**Variables**	**Odds Ratio (95% CI)**	** *p* ** **-Value**
Age for every 1-year increase	1.023 (1.007–1.039)	0.004
Smoking	2.247 (1.068–4.728)	0.033
Coronary heart disease	2.709 (1.250–5.873)	0.012
Wheezing	3.128 (1.109–8.824)	0.031
Runny nose	1.911 (1.050–3.477)	0.034
Previous treatment of dyspnea	6.749 (3.670–12.409)	<0.001

**Table 4 diseases-13-00130-t004:** Clinical and pulmonary function data of patients with COPD and asthma.

**Variables**	**COPD** **(n = 84)**	**Asthma** **(n = 104)**	** *p* ** **-Value**
Age, years	60.9 ± 13.8	61.2 ± 15.4	0.880
Female	42 (50.0)	78 (75.0)	<0.001
Male	42 (50.0)	26 (25.0)	<0.001
Body mass index, kg/m^2^	24.6 ± 4.6	24.4 ± 4.9	0.803
Smoking (current or former)	19 (22.6)	7 (6.7)	<0.001
Amount of smoking, pack-years	18.3 ± 19.3	14.6 ± 26.8	0.635
Fuel, hours per year	9666.7 ± 13,625.2	10,434.7 ± 8232.5	0.667
**Occupations**			
Government officer	8 (9.5)	8 (7.7)	0.655
Farmer	12 (11.5)	4 (4.8)	0.098
Others	30 (35.7)	29 (27.9)	0.250
Unemployed	41 (48.8)	55 (52.9)	0.480
**Preexisting comorbidities**			
Hypertension	31 (36.9)	45 (43.3)	0.377
Hyperlipidemia	22 (26.2)	38 (36.5)	0.130
Diabetes	15 (17.9)	18 (17.3)	0.922
Coronary heart disease	8 (9.5)	9 (8.7)	0.836
Cerebrovascular disease	0 (0)	3 (2.9)	0.167
Obesity	1 (1.2)	7 (6.7)	0.061
Allergic rhinitis	12 (14.3)	12 (11.5)	0.575
Asthma	2 (2.4)	25 (24.0)	<0.001
COPD	8 (9.5)	1 (1.0)	0.006
**Respiratory symptoms**	41 (48.8)	58 (55.8)	0.342
Cough	21 (25.0)	33 (31.7)	0.311
Sputum production	17 (20.2)	20 (19.2)	0.863
Breathlessness	11 (13.1)	26 (25.0)	0.041
Wheezes	5 (6.0)	8 (7.7)	0.640
Chest tightness	3 (3.6)	5 (4.8)	0.733
Runny nose	9 (10.7)	16 (15.4)	0.348
Nasal obstruction	7 (8.3)	8 (7.7)	0.872
Sore throat	4 (4.8)	4 (3.8)	0.757
Dyspnea on exertion	3 (3.6)	7 (6.7)	0.516
**History of respiratory treatment and cost**			
Previous treatment of dyspnea	14 (16.7)	23 (22.1)	0.350
Prior ED visit in the past year	4 (4.8)	8 (7.7)	0.414
**Spirometry data**			
FVC, L	2.86 ± 0.88	2.15 ± 0.67	<0.001
FVC, %predicted	100.4 ± 19.0	85.7 ± 19.8	<0.001
FVC improvement after BDR test, %	−0.1 ± 6.9	12.8 ± 12.1	<0.001
FEV_1_, L	1.95 ± 0.63	1.65 ± 0.51	0.001
FEV_1_, %predicted	84.7 ± 17.1	81.7 ± 20.8	0.027
FEV_1_ improvement after BDR test, %	4.3 ± 4.8	15.0 ± 10.9	<0.001
FEV_1_/FVC, %	68.2 ± 6.3	77.9 ± 11.4	<0.001
FEF_25-75_, L/s	1.10 ± 0.54	1.46 ± 0.89	0.001
FEF_25-75_, %predicted	44.1 ± 15.2	65.0 ± 35.2	<0.001
BDR	5 (6.0)	85 (81.7)	<0.001

Data shown as n (%) or mean ± SD. BDR defined as increase in FEV_1_ ≥ 12% and ≥200 mL after BDR test. BDR = bronchodilator response, COPD = chronic obstructive pulmonary disease, ED = emergency department, FEV_1_ = forced expiratory volume in one second, FVC = forced vital capacity, FEF_25-75_ = forced expiratory flow at 25–75% of FVC, kg = kilogram, L = liter, m = meter, mL = milliliter, s = second.

## Data Availability

Data are contained within the article.

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
