# Peer review of "Prevalence of Chronic Obstructive Pulmonary Disease and Asthma in the Community of Pathumthani, Thailand"

_diseases, 2025, doi:10.3390/diseases13050130_

Round 1
Reviewer 1 Report
Comments and Suggestions for Authors
This MS provides valuable epidemiological data on airway diseases in Thailand, emphasizing the importance of spirometry in community-based disease surveillance. However, some methodological and analytical limitations need to be addressed.
1. The title is clear and accurately reflects the study's focus, but could be slightly more concise and impactful; suggested title: "Prevalence and Risk Factors of COPD and Asthma in a Thai Community: A Cross-Sectional Study."
2. Include effect sizes (Odds Ratios, Confidence Intervals) for key associations. Improve clarity and specificity in the abstract.
3. Clearly state the hypothesis and expected findings in the introduction section.
4. No mention of sample size calculation details.
5. How were confounders controlled in the logistic regression model?
6. No details on intra-observer variability or quality control in spirometry.
7. Effect sizes for logistic regression findings are missing in the main text (Odds Ratios should be emphasized).
8. Table 4 is not necessary.
9. Avoid over-speculation; the authors should cite studies linking PM2.5 to COPD prevalence changes.
10. Address potential study biases and their impact.
11. No discussion of study limitations.
Language and grammar need revision for clarity and readability.
Author Response
Comment 1: The title is clear and accurately reflects the study's focus, but could be slightly more concise and impactful; suggested title: "Prevalence and Risk Factors of COPD and Asthma in a Thai Community: A Cross-Sectional Study.
Response 1: Thank you very much for your suggestion regarding the study title. However, we are unable to change the title due to ethical and funding constraints.
Comment 2: Include effect sizes (Odds Ratios, Confidence Intervals) for key associations. Improve clarity and specificity in the abstract.
Response 2: We added the odds ratios and confidence intervals to the abstract (page 1, lines 32-36).
Comment 3: Clearly state the hypothesis and expected findings in the introduction section.
Response 3: We added the hypothesis and expected findings in the introduction section (page 2, lines 67-73).
Comment 4: No mention of sample size calculation details.
Response 4: We have provided the sample size calculation in the Statistical Analysis section (page 3, lines 114-118). Thank you very much.
Comment 5: How were confounders controlled in the logistic regression model?
Response 5: An explanation of how confounders were controlled has been added to the Statistical Analysis section (page 3, lines 127-131).
Comment 6: No details on intra-observer variability or quality control in spirometry.
Response 6: We added additional details to the Procedures and Outcomes section (page 3, lines 96-100).
Comment 7: Effect sizes for logistic regression findings are missing in the main text (Odds Ratios should be emphasized)
Response 7: We added the effect sizes to the main text (page 7, lines 165-169).
Comment 8: Table 4 is not necessary.
Response 8: We believe it is important to retain Table 4, as it allows us to demonstrate and discuss the differences between COPD and asthma. Thank you for your feedback.
Comment 9: Avoid over-speculation; the authors should cite studies linking PM2.5 to COPD prevalence changes.
Response 9: We have cited relevant studies linking PM2.5 to COPD prevalence in the Discussion section (page 9, lines 195-200).
Comment 10: Address potential study biases and their impact.
Response 10: We have added a discussion of potential study biases and their impact in the Discussion section (page 10, line 261-266).
Comment 11: No discussion of study limitations.
Response 11: We have added a discussion of study limitations in the Discussion section (page 10, lines 249-271).

Reviewer 2 Report
Comments and Suggestions for Authors
This is a very interesting population-based study about the prevalence of asthma and COPD in the Community of Pathumthani, Thailand.
It clearly shows that both COPD and asthma rates have been increased compared to the previous similar surveys in Thailand.
I suggest the authors to pay more attention in the discussion part about the possible role of occupational exposure (current or previous) on the results, maybe with comparisons with similar studies in Thailand or its neighboring countries, or perhaps globally.
It would be nice, if it is possible, to divide participants (included in "others") by exposure assessment (not only farmers and government officers), perhaps into groups of exposed to VGDF and not exposed, or similar...).
If this is applicable for authors, I think it will significantly improve the scientific value of the manuscript.
Author Response
Comments 1: This is a very interesting population-based study about the prevalence of asthma and COPD in the Community of Pathumthani, Thailand.
It clearly shows that both COPD and asthma rates have been increased compared to the previous similar surveys in Thailand.
Response 1: We would like to express my heartfelt gratitude to the reviewer for the wonderful reviews and comments.
Comments 2: I suggest the authors to pay more attention in the discussion part about the possible role of occupational exposure (current or previous) on the results, maybe with comparisons with similar studies in Thailand or its neighboring countries, or perhaps globally.
Response 2: We have added a discussion of the possible role of occupational exposure in the Discussion section (page 6, lines 211-213), including relevant comparisons where applicable.
Comments 3: It would be nice, if it is possible, to divide participants (included in "others") by exposure assessment (not only farmers and government officers), perhaps into groups of exposed to VGDF and not exposed, or similar...). If this is applicable for authors, I think it will significantly improve the scientific value of the manuscript.
Response 3: We have categorized the “other” occupations into merchant, general worker, and others (Table 1). However, we are unable to further stratify participants into VGDF-exposed and non-exposed groups due to insufficient data. Thank you very much for your valuable suggestions.

Reviewer 3 Report
Comments and Suggestions for Authors
Dear authors,
I read with interest your paper on airway disease prevalence in Pathumthani community. Here are my main comments:
- The novelty of the study should be more specifically addressed in the introduction and in discussion section of the study.
- Where there patients with persistent airflow limitation in the enrolled cohort? How were they considered? What about patients with strong clinical suspicion of asthma but normal spirometry and negative BDR? Were they assessed with bronchial challenge test?
- In sample size calculation, what does “102” refer to? Asthmatic patients? COPD? Overall cohort? Please explain
- Non significant associations in univariate regression should be displayed at least in a supplementary document. Did the authors evaluated the overall robustness of the model? From table 3 I cannot retrieve this information
- Table 2 is not so informative in the proposed structure. I believe it could be better adding a table comparing healthy subjects, asthmatic and COPD lung function.
- There are no info regarding disease severity (exacerbations, ACT-ACQ, CAT-mMRC etc) nor prescribed treatment.
- CV comorbidities did not differ between asthma and COPD patients. Given that patients with COPD usually show a higher degree of CV damage, how would you explain these results?
Author Response
Comments 1: The novelty of the study should be more specifically addressed in the introduction and in discussion section of the study.
Response 1: We have added the novelty of the study more specifically in both the Introduction and the Discussion sections (page 2 lines 67-73; and page 8 lines 187-190).
Comments 2: Where there patients with persistent airflow limitation in the enrolled cohort? How were they considered? What about patients with strong clinical suspicion of asthma but normal spirometry and negative BDR? Were they assessed with bronchial challenge test?
Response 2: We have addressed these issues in the limitation of study in the Discussion section (page 10 lines 255-261).
Comments 3: In sample size calculation, what does “102” refer to? Asthmatic patients? COPD? Overall cohort? Please explain.
Response 3: We have clarified this issue in the Statistical Analysis section (page 3 lines 117-118).
Comments 4: Non significant associations in univariate regression should be displayed at least in a supplementary document. Did the authors evaluated the overall robustness of the model? From table 3 I cannot retrieve this information.
Response 4: We have included the results of the univariate analysis in a supplementary table. Additionally, I assessed the overall robustness of the model and have now specified the use of the backward stepwise selection method in the Statistical Analysis section (page 3 lines 131-132).
Comments 5: Table 2 is not so informative in the proposed structure. I believe it could be better adding a table comparing healthy subjects, asthmatic and COPD lung function.
Response 5: I have revised Table 2 to include a comparison of lung function among healthy subjects, asthmatic patients, and those with COPD (Table 2 page 5).
Comments 6: There are no info regarding disease severity (exacerbations, ACT-ACQ, CAT-mMRC etc) nor prescribed treatment.
Response 6: We are unable to report the severity of airway diseases or their treatment due to insufficient data. However, I have addressed these limitations in the Discussion section (page 10 lines 266-267).
Comments 7: CV comorbidities did not differ between asthma and COPD patients. Given that patients with COPD usually show a higher degree of CV damage, how would you explain these results?
Response 7: We have addressed this issue in the Discussion section (page 9 lines 226-233).

Round 2
Reviewer 1 Report
Comments and Suggestions for Authors
The clinical characteristics and pulmonary function profiles of patients with COPD and asthma have been extensively studied. Given that this study specifically undertook a comparative analysis, it is necessary to discuss the differences between the patient population in this region and those reported in previous studies.
Author Response
Comments 1: The clinical characteristics and pulmonary function profiles of patients with COPD and asthma have been extensively studied. Given that this study specifically undertook a comparative analysis, it is necessary to discuss the differences between the patient population in this region and those reported in previous studies.
Response 1: We have added a discussion of the differences between the patient population in this region and those reported in previous studies in the Discussion section (page 9, lines 194–212). Thank you very much.

Reviewer 3 Report
Comments and Suggestions for Authors
Dear authors,
thanks for the revised paper. I have no further request.
Author Response
Comments 1: Dear authors,
Thanks for the revised paper. I have no further request.
Response 1: I truly appreciate the reviewer’s thoughtful and encouraging comments.
Round 3
Reviewer 1 Report
Comments and Suggestions for Authors
Accept.